# METAGROSS: META GATED RECURSIVE CONTROLLER UNITS FOR SEQUENCE MODELING

## ABSTRACT

This paper proposes METAGROSS (Meta Gated Recursive Controller), a new neural sequence modeling unit. Our proposed unit is characterized by recursive parameterization of its gating functions, i.e., gating mechanisms of METAGROSS are controlled by instances of itself, which are repeatedly called in a recursive fashion. This can be interpreted as a form of meta-gating and recursively parameterizing a recurrent model. We postulate that our proposed inductive bias provides modeling benefits pertaining to learning with inherently hierarchically-structured sequence data (e.g., language, logical or music tasks). To this end, we conduct extensive experiments on recursive logic tasks (sorting, tree traversal, logical inference), sequential pixel-by-pixel classification, semantic parsing, code generation, machine translation and polyphonic music modeling, demonstrating the widespread utility of the proposed approach, i.e., achieving state-of-the-art (or close) performance on all tasks.

## 1 INTRODUCTION

Sequences are fundamentally native to the world we live in, i.e., language, logic, music and time are all well expressed in sequential form. To this end, the design of effective and powerful sequential inductive biases has far-reaching benefits across many applications. Across many of these domains, e.g., natural language processing or speech, the sequence encoder lives at the heart of many powerful state-of-the-art model architectures.

Models based on the notion of recurrence have enjoyed pervasive impact across many applications. In particular, the best recurrent models operate with gating functions that not only ameliorate vanishing gradient issues but also enjoy fine-grain control over temporal compositionality (Hochreiter & Schmidhuber, 1997; Cho et al., 2014). Specifically, these gating functions are typically static and trained via an alternate transformation over the original input.

In this paper, we propose a new sequence model that recursively parameterizes the recurrent unit. More concretely, the gating functions of our model are now parameterized repeatedly by instances of itself which imbues our model with the ability to reason deeply[1] and recursively about certain inputs. To achieve the latter, we propose a soft dynamic recursion mechanism, which softly learns the depth of recursive parameterization at a per-token basis. Our formulation can be interpreted as a form of meta-gating since temporal compositionality is now being meta-controlled at various levels of abstractions.

Our proposed method, Meta Gated Recursive Controller Units (METAGROSS), marries the benefits of recursive reasoning with recurrent models. Notably, we postulate that this formulation brings about benefits pertaining to modeling data that is instrinsically hierarchical (recursive) in nature, e.g., natural language, music and logic, an increasingly prosperous and emerging area of research (Shen et al., 2018; Wang et al., 2019; Choi et al., 2018). While the notion of recursive neural networks is not new, our work is neither concerned with syntax-guided composition (Tai et al., 2015; Socher et al., 2013; Dyer et al., 2016) nor unsupervised grammar induction (Shen et al., 2017; Choi et al., 2018; Havrylov et al., 2019; Yogatama et al., 2016). Instead, our work is a propulsion on a different frontier, i.e., learning recursively parameterized models which bears a totally different meaning.

---

[1]This is meant literally.

Overall, the key contributions of this work are as follows:

- We propose a new sequence model. Our model is distinctly characterized by recursive parameterization of recurrent gates, i.e., compositional flow is controlled by instances of itself, á la repeatedly and recursively. We propose a soft dynamic recursion mechanism that dynamically and softly learns the recursive depth of the model at a token-level.

- We propose a non-autoregressive parallel variation of METAGROSS,that when equipped with the standard Transformer model (Vaswani et al., 2017), leads to gains in performance.

- We evaluate our proposed method on a potpourri of sequence modeling tasks, i.e., logical recursive tasks (sorting, tree traversal, logical inference), pixel-wise sequential image classification, semantic parsing, neural machine translation and polyphonic music modeling. METAGROSS achieves state-of-the-art performance (or close) on all tasks.

## 2 META GATED RECURSIVE CONTROLLERS (METAGROSS)

This section introduces our proposed model. METAGROSS is fundamentally a recurrent model. Our proposed model accepts a sequence of vectors $X \in \mathbb{R}^{\ell \times d}$ as input. The main unit of the Metagross unit $h_t = \text{Metagross}_n(x_t, h_{t-1})$ is defined as follows:

$$f_t^n = \sigma_s(\alpha_t \, \text{Metagross}_{n+1}(x_t, h_{t-1}) + (1 - \alpha_t) \, F_n^F(x_t, h_{t-1}))$$
$$o_t^n = \sigma_s(\beta_t \, \text{Metagross}_{n+1}(x_t, h_{t-1}^n) + (1 - \beta_t) \, F_n^O(x_t, h_{t-1}))$$
$$z_t^n = \sigma_r(F_n^Z(x_t, h_t))$$
$$c_t^n = (1 - f_t) \odot h_{t-1} + (f_t) \odot z_t$$
$$h_t^n = o_t \odot c_t$$
$$h_t^n = h_t + x_t$$

where $\sigma_r$ is a nonlinear activation such as tanh. $\sigma_s$ is the sigmoid activation function. In a nutshell, the Metagross unit recursively calls itself until a max depth $L$ is hit. When $n = L$, $f_t$ and $o_t$ are parameterized by:

$$f_t^L = \sigma_s(F_L^F(x_t, h_{t-1}))$$
$$o_t^L = \sigma_s(F_L^O(x_t, h_{t-1}))$$

where $f_t^L, o_t^L$ is the forget and output gate of METAGROSS at time step $t$ while at the maximum depth $L$. We also include an optional residual connection $h_t^n = h_t + x_t$ to facilitate gradient flow down the recursive parameterization of METAGROSS.

### 2.1 SOFT DYNAMIC RECURSION

We propose learning the depth of recursion in a data-driven fashion. To learn $\alpha_t, \beta_t$, we use the following:

$$\alpha_t = F_\alpha(x_t) \text{ and } \beta_t = F_\beta(x_t)$$

where $F_*(xt) = W x_t + b$ is a simple linear transformation layer applied to sequence X across the temporal dimension. Intuitively, $\alpha, \beta$ control the extent of recursion, enabling a *soft* depth pertaining to the hierarchical parameterization. Alternatively, we may also consider a static variation where:

$$\alpha_t = F_\alpha(\sum_{t=0}^{\ell} x_t) \text{ and } \beta_t = F_\beta(\sum_{t=0}^{\ell} x_t)$$

where the same value of $\alpha, \beta$ is computed based on global information from the entire sequence. Note that this strictly cannot be used for autoregressive decoding. Finally, we note that it is also possible to assign $\alpha \in \mathbb{R}, \beta \in \mathbb{R}$ to be trainable scalar parameters.

## 2.2 LEVEL-WISE PARAMETERIZATION

Intuitively, $F_n^* \forall * \in F, O, Z$ are level-wise parameters of METAGROSS. We parameterize $F_n$ with either level-wise RNN units or simple linear transformations.

$$F_n^*(x_t) = \text{RNN}_n^*(x_t, h_{t-1}^n) \quad \textbf{or} \quad F_n^*(x_t) = W_n^* x_t + b_n^*$$

Overall, METAGROSS is agnostic to the choice of $F_n^*(x_t)$ and even the RNN unit. Note that for RNN, the hidden states are initialized with zero and each level uses a new initial hidden state.

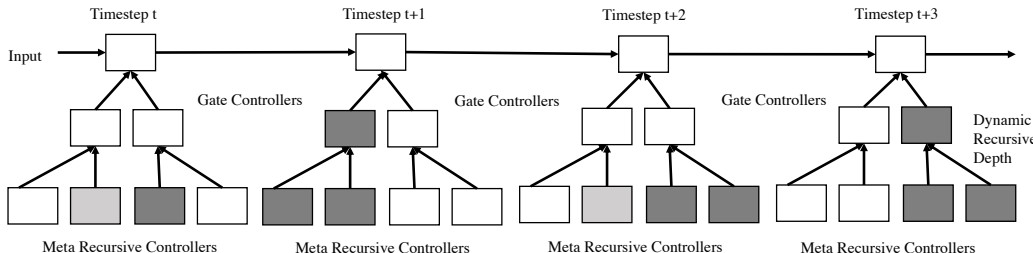

Figure 1: Architecture of Proposed METAGROSS network with maximum depth $N = 3$. The extent of greyness illustrates effect from soft dynamic recursion.

## 2.3 PARALLEL NON-AUTOREGRESSIVE VARIATION

We postulate that METAGROSS can also be useful as a non-autoregressive parallel model. This can be interpreted as a form of recursive feed-forward layer that is used in place of recurrent META-GROSS for speed benefits. In early experiments, we find this a useful enhancement to state-of-the-art Transformer (Vaswani et al., 2017) models. The non-autoregressive variant of METAGROSS is written as follows:

$$f_t^n = \sigma_s(\alpha_t \, \text{Metagross}_{n+1}(x_t) + (1 - \alpha_t) \, F_n^F(x_t))$$
$$o_t^n = \sigma_s(\beta_t \, \text{Metagross}_{n+1}(x_t) + (1 - \beta_t) \, F_n^O(x_t))$$
$$z_t^n = \sigma_r(F_n^Z(x_t))$$
$$h_t^n = (f_t^n \odot x_t) + (o_t \odot z_t^n)$$
$$h_t^n = h_t + x_t$$

More concretely, we dispense with the reliance on the previous hidden state. This can be used in place of any position-wise feed-forward layer. In this case, note that $F_n^*(x_t)$ are typically position-wise functions as well.

## 3 EXPERIMENTS

We conduct experiments on a suite of diagnostic synthetic tasks and real world tasks.

## 3.1 LOGIC AND RECURSIVE TASKS

We evaluate our model on three diagnostic logical tasks as follows:

- **Task 1** (SORT SEQUENCES) - The input to the model is a sequence of integers. The correct output is the sorted sequence of integers. Since mapping sorted inputs to outputs can be implemented in a recursive fashion, we evaluate our model's ability to better model recursively structured sequence data. Example input output pair would be $9, 1, 10, 5, 3 \rightarrow 1, 3, 5, 9, 10$.

- **Task 2** (TREE TRAVERSAL) - We construct a binary tree of maximum depth $N$. The goal is to generate the *postorder* tree traversal given the *inorder* **and** *preorder* traversal of the tree. Note that this is known to arrive at only one unique solution. The constructed trees have random sparsity where we assign a probability $p$ of growing the tree up to maximum

depth $N$. Hence, the trees can be of varying depths[2]. This requires inferring hierarchical structure and long-term reasoning across sequences. We concatenate the *postorder* and *inorder* sequences, delimiteted by a special token. We evaluate on $n \in \{3, 4, 5, 8, 10\}$. For $n = \{5, 8\}$, we ensure that each tree traversal has at least 10 tokens. For $n = 10$, we ensure that each path has at least 15 tokens. Example input output pair would be $13, 15, 4, 7, 5, X, 13, 4, 15, 5, 7 \rightarrow 7, 15, 13, 4, 5$.

- **Task 3** (LOGICAL INFERENCE) - We use the standard logical inference dataset[3] proposed in (Bowman et al., 2014). This is a classification task in which the goal is to determine the semantic equivalence of two statements expressed with logic operators such as *not*, *and*, and *or*. The language vocabulary is of six words and three logic operators. As per prior work (Shen et al., 2018), the model is trained on sequences with 6 or less operations and evaluated on sequences of 6 to 12 operations.

For Task 1 and Task 2, we frame these tasks as a Seq2Seq (Sutskever et al., 2014) task and evaluate models on exact match accuracy and perplexity (P) metrics. We use a standard encoder-decoder architecture with attention (Bahdanau et al., 2014). We vary the encoder module with BiLSTMs, Stacked BiLSTMs (3 layers) and Ordered Neuron LSTMs (Shen et al., 2018). For Task 3 (logical inference), we use the common setting in other published works.

**Results on Sorting and Tree Traversal** Table 1 reports our results on the Sorting and Tree Traversal task. All models solve the task with $n = 3$. However, the task gets increasingly harder with a greater maximum possible length and largely still remains a challenge for neural models today. The relative performance of METAGROSS is on a whole better than any of the baselines, especially pertaining to perplexity. We also found that S-BiLSTMs are always better than LSTMs on this task and Ordered LSTMs are slightly worst than vanilla BiLSTMs. However, on sorting, ON-LSTMs are much better than standard BiLSTMs and S-BiLSTMs.

| | TREE TRAVERSAL | | | | | | | | | | SORT | | | |
| | $n = 3$ | | $n = 4$ | | $n = 5$ | | $n = 8$ | | $n = 10$ | | $n = 5$ | | $n = 10$ | |
| Model | EM | P | EM | P | EM | P | EM | P | EM | P | EM | P | EM | P |
|---|---|---|---|---|---|---|---|---|---|---|---|---|---|---|
| BiLSTM | 100 | 1.0 | 96.9 | 2.4 | 60.3 | 2.4 | 5.6 | 30.6 | 2.2 | 132 | 79.9 | 1.2 | 78.9 | 1.2 |
| S-BiLSTM | 100 | 1.0 | 98.0 | 1.0 | 63.4 | 2.5 | **5.9** | 99.9 | 2.8 | 225 | 83.4 | 1.2 | 88.0 | 1.1 |
| ON-LSTM | 100 | 1.0 | 81.0 | 1.4 | 55.7 | 2.8 | 5.5 | 52.3 | 2.7 | 173 | 90.8 | 1.1 | 87.4 | 1.1 |
| METAGROSS | **100** | **1.0** | **98.4** | **1.0** | **63.4** | **1.8** | 5.6 | **20.4** | **2.8** | **119** | **92.2** | **1.1** | **90.6** | **1.1** |

Table 1: Experimental Results on Tree Traversal and Sorting.

| | # Operations | | | | | |
| Model | 7 | 8 | 9 | 10 | 11 | 12 |
|---|---|---|---|---|---|---|
| Tree-LSTM[†] (Tai et al., 2015) | 93 | 90 | 87 | 89 | 86 | 87 |
| LSTM (Bowman et al., 2014) | 88 | 85 | 80 | 78 | 71 | 69 |
| RRNet (Jacob et al., 2018) | 84 | 81 | 78 | 74 | 72 | 71 |
| ON-LSTM (Shen et al., 2018) | 91 | 87 | 86 | 81 | 78 | 76 |
| METAGROSS | **97** | **95** | **93** | **92** | **90** | **88** |

Table 2: Experimental results on Logical Inference task. † denotes models with access to ground truth syntax. Results reported from (Shen et al., 2018). METAGROSS achieves state-of-the-art performance.

**Results on Logical Inference** Table 2 reports our results on logical inference task. We compare with mainly other published work. METAGROSS is a strong and competitive model on this task, outperforming ON-LSTM by a wide margin ($+12\%$ on the longest nunber of operations). Performance of our model also exceeds Tree-LSTM, which has access to ground truth syntax. Our model achieves state-of-the-art performance on this dataset even when considering models with access to syntactic information.

---

[2]Note that all our models solves the problem entirely when the tree is fixed and full. Hence, random trees provide a necessary challenge.

[3]3https://github.com/sleepinyourhat/vector-entailment.

## 3.2 Pixel-wise Sequential Image Classification

We evaluate our model on its ability to model and capture long-range dependencies. More specifically, the sequential pixel-wise image classification problem treats pixels in images as sequences. We use the well-established pixel-wise MNIST and CIFAR-10 datasets. We use 3 layered META-GROSS of 128 hidden units each.

| Model | $|\theta|$ | MNIST | CIFAR |
|---|---|---|---|
| DilatedGRU (Chang et al., 2017) | - | 99.00 | - |
| IndRNN (Li et al., 2018a) | - | 99.00 | - |
| r-LSTM (Trinh et al., 2018) | - | 98.52 | 72.20 |
| Transformer (Trinh et al., 2018) | - | 98.90 | 62.20 |
| TrellisNet (Bai et al., 2018b) | 8.0M | **99.20** | 73.42 |
| TrellisNet (Our run) | 8.0M | 97.59 | 55.83 |
| METAGROSS | 0.9M | 99.04 | 73.01 |
| METAGROSS$^\sharp$ | 0.9M | 99.09 | **73.95** |

Table 3: Experimental results (accuracy) on Pixel-wise Sequential Image Classification. We also trained the recent R-Adam optimizer (Liu et al., 2019) which we found to have improved performance (results denoted with $\sharp$).

**Results on Pixel-wise Image Classification** Table 3 reports the results of METAGROSS against other published works. Our method achieves state-of-the-art performance on the CIFAR-10 dataset, outperforming the recent Trellis Network (Bai et al., 2018b). On the other hand, results on MNIST are reasonable, outperforming a wide range of other published works. On top of that, our method has 8 times less parameters than Trellis network (Bai et al., 2018b) while achieving similar or better performance. This ascertains that METAGROSS is a reasonably competitive long-range sequence encoder.

## 3.3 Semantic Parsing / Code Generation

We evaluate METAGROSS on semantic parsing (GEO, ATIS, JOBS) and code generation (DJANGO). We run our experiments on the publicly released code[4] of (Yin & Neubig, 2018), replacing the recurrent decoder with our METAGROSS decoder. Hyperparameter details followed the codebase of (Yin & Neubig, 2018) quite strictly.

| Model | GEO | ATIS | JOBS | DJANGO |
|---|---|---|---|---|
| Seq2Tree (Dong & Lapata, 2016) | 87.1 | 84.6 | - | 31.5 |
| LPN (Ling et al., 2016) | - | - | - | 62.3 |
| NMT (Neubig, 2015) | - | - | - | 45.1 |
| YN17 (Yin & Neubig, 2017) | - | - | - | 71.6 |
| ASN (Rabinovich et al., 2017) | 85.7 | 85.3 | - | - |
| ASN+Att (Rabinovich et al., 2017) | 87.1 | 85.9 | - | - |
| TranX (Yin & Neubig, 2018) | 88.2 | 86.2 | - | 72.7 |
| TranX (Code reported) | **88.6** | 87.7 | 90.0 | 77.2 |
| TranX (Our Run) | 87.5 | 87.5 | 90.0 | 76.7 |
| TranX + METAGROSS | **88.6** | **88.4** | **90.7** | **78.3** |

Table 4: Experimental results on Semantic Parsing and Code Generation.

**Results on Semantic Parsing and Code Generation** Table 4 reports our experimental results on Semantic Parsing (GEO, ATIS, JOBS) and Code Generation (DJANGO). We observe that TranX + METAGROSS outperforms all competitor approaches, achieving state-of-the-art performance. More importantly, the performance gain over the base TranX method allows us to observe the ablative benefits of METAGROSS.

## 3.4 Neural Machine Translation

We conduct experiments on two IWSLT datasets which are collections derived from TED talks. Specifically, we compare on the IWSLT 2014 German-English and IWSLT 2015 English-Vietnamese datasets. We compare against a suite of published results and strong baselines. For our method, we replaced the multi-head aggregation layer in the Transformer networks (Vaswani et al., 2017) with a parallel non-autoregressive adaptation of METAGROSS. The base models are all linear layers. For our experiments, we use the standard implementation and hyperparameters in Tensor2Tensor[5] (Vaswani et al., 2018), using the small (S) and base (B) setting for Transformers. Model averaging is used and beam size of $8/4$ and length penalty of $0.6$ is adopted for De-En and En-Vi respectively. For our model, max depth is tuned amongst $\{1, 2, 3\}$. We also ensure to compare, in an ablative fashion, our own reported runs of the base Transformer models.

---

[4] https://github.com/pcyin/tranX
[5] https://github.com/tensorflow/tensor2tensor

| Model | BLEU |
|---|---|
| MIXER (Ranzato et al., 2015) | 21.83 |
| AC+LL (Bahdanau et al., 2016) | 28.53 |
| NPMT (Huang et al., 2017) | 28.96 |
| Dual Transfer (Wang et al., 2018) | 32.35 |
| Transformer S (Vaswani et al., 2017) | 32.86 |
| Layer-wise (He et al., 2018) | 35.07 |
| Transformer S (Our run) | 34.68 |
| Transformer B (Our run) | 36.30 |
| Transformer S + METAGROSS | 35.15 |
| Transformer B + METAGROSS | **37.09** |

Table 5: Experimental results on Neural Machine Translation on IWSLT 2014 De-En.

| Model | BLEU |
|---|---|
| (Luong & Manning, 2015) | 23.30 |
| Att-Seq2Seq (Bahdanau et al., 2014) | 26.10 |
| NPMT (Huang et al., 2017) | 27.69 |
| NPMT + LM (Huang et al., 2017) | 28.07 |
| Transformer B (Vaswani et al., 2017) | 28.43 |
| Transformer B + METAGROSS | **30.81** |

Table 6: Experimental results on Neural Machine Translation on IWSLT 2015 En-Vi.

**Results on Neural Machine Translation** Table 5 reports results on IWSLT 2014 de-en task. Our proposed model performs very competitively (37.09 BLEU), outperforming many well-established baselines. Our results also show that equipping Transformer models with METAGROSS can also lead to improvements in performance. Notably there is a +0.69 BLEU improvement on Transformer Base and +0.42 BLEU improvement for Transformer Small. On the other hand, our method achieves 30.81 BLEU on the IWSLT 2015 En-Vi dataset, with +2.38 improvement in BLEU from the standard Transformer Base model.

## 3.5 POLYPHONIC MUSIC MODELING

We evaluate METAGROSS on the polyphonic music modeling. We use three well-established datasets, namely Nottingham, JSB Chorales and Piano Midi (Boulanger-Lewandowski et al., 2012). The input to the model are 88-bit sequences, each corresponding to the 88 keys of the piano. The task is evaluated on the Negative Log-likelihood (NLL). We compare with a wide range of published works (Chung et al., 2014; Bai et al., 2018a; Song et al., 2019)

| Model | Nott | JSB | Piano |
|---|---|---|---|
| GRU (Chung et al.) | 3.13 | 8.54 | 8.82 |
| LSTM (Song et al.) | 3.25 | 8.61 | 7.99 |
| G2-LSTM (Li et al.) | 3.21 | 8.67 | 8.18 |
| B-LSTM (Song et al.) | 3.16 | 8.30 | 7.55 |
| TCN (Bai et al.) | 3.07 | **8.10** | - |
| TCN (our run) | 2.95 | 8.13 | 7.53 |
| METAGROSS | **2.88** | 8.12 | **7.49** |

Table 7: Experimental Results (NLL) on Polyphonic Music Modeling.

**Results on Music Modeling** Table 7 reports our scores on this task. METAGROSS achieves state-of-the-art performance on the Nottingham and Piano midi datasets, outperforming a wide range of competitive models such as Gumbel Gate LSTMs (Li et al., 2018b).

## 3.6 ANALYSIS AND DISCUSSION

This section reports some analysis and discussion regarding the proposed model.

### 3.6.1 EFFECT OF MAXIMUM DEPTH AND BASE UNIT

| Max $N$ | Base Model | ATIS | DJANGO |
|---|---|---|---|
| 2 | Linear | **88.40** | 77.56 |
| 3 | Linear | 88.21 | 77.62 |
| 4 | Linear | 87.80 | 76.84 |
| 2 | LSTM | 86.61 | **78.33** |
| 3 | LSTM | 85.93 | 77.39 |

Table 8: Ablation studies on Semantic Parsing and Code Generation.

| Task | $N$ | Base Unit |
|---|---|---|
| Tree Traversal | 2 | Recurrent |
| Sorting | 2 | Recurrent |
| Logical Inference | 3 | Recurrent |
| Pixel-wise Classification | 2 | Recurrent |
| Semantic Parsing | 2 | Linear |
| Code Generation | 2 | Recurrent |
| Machine Translation | 3 | Linear |
| Polyphonic Music | 3 | Linear |

Table 9: Optimal Maximum Depth $N$ and base unit for different tasks.

Table 8 reports some ablation studies on the semantic parsing and code generation tasks. We observe that the base unit and optimal maximum depth used is task dependent. For ATIS dataset, using the linear transform as the base unit performs the best. Conversely, the linear base unit performs worse than the recurrent base unit (LSTM) on the DJANGO dataset.

On a whole, we also observed this across other tasks, i.e., the base unit and maximum depth of METAGROSS is a critical choice for most tasks. Table 9 reports the optimal max depth $N$ and best base unit for each task.

### 3.6.2 ANALYSIS OF SOFT DYNAMIC RECURSION

Figure 6 illustrates the depth gate values on CIFAR and MNIST datasets. These values reflect the $\alpha$ and $\beta$ values in METAGROSS, signifying how the parameter tree is being constructed during training. This is reflected as L and R in the figures representing left and right gates. Firstly, we observe that our model indeed builds data-specific parameterization of the network. This is denoted by how METAGROSS builds different[6] trees for CIFAR and MNIST.

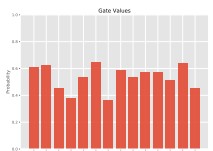
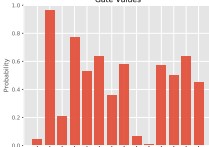
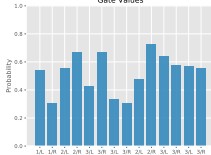
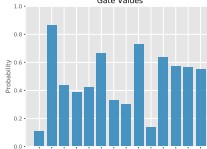

Figure 2: Depth Gates at Initial (CIFAR)

Figure 3: Depth Gates at epoch 10 (CIFAR)

Figure 4: Depth Gates at Initial (MNIST)

Figure 5: Depth Gates at epoch 10 (MNIST)

Figure 6: Depth Gate Visualization on CIFAR and MNIST.

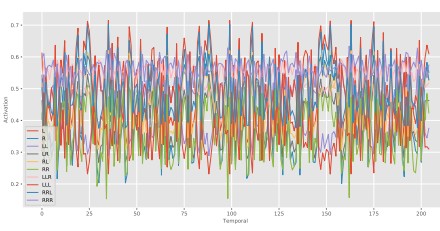

Figure 7: Dynamic Recursion on CIFAR.

Secondly, we analyze the dynamic recursion depth with respect to time steps. The key observation that all datasets have very diverse construction of recursive parameters. The recursive gates fluctuate aggressively on CIFAR while remaining more stable on Music modeling. Moreover, we found that the recursive gates remain totally constant on MNIST. This demonstrates that our model has the ability to adjust the dynamic construction adaptively and can revert to static recursion over time if necessary. We find that compelling.

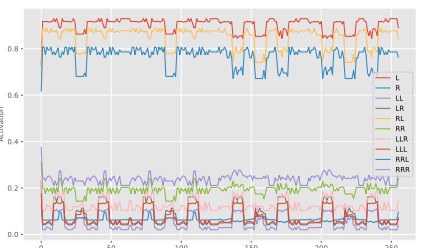

Figure 8: Dynamic Recursion on Music.

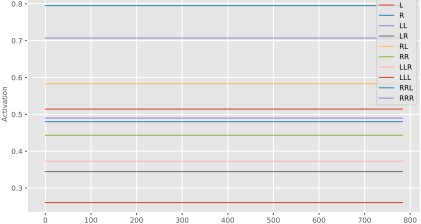

Figure 9: Dynamic Recursion on MNIST.

---

[6]Though not depicted, we also found that the probability of each node expanding to children has low variance across batches in the same dataset.

The adaptive recursive depth is made more intriguing by observing how the recursive parameterization alters on CIFAR and Music datasets. From Figure 8 we observe that the structure of the network changes in a rhythmic fashion, in line with our intuition of musical data. When dealing with pixel information, the tree structure changes adaptively according to the more complex information processed by the network.

## 4 RELATED WORK

The study of effective inductive biases for sequential representation learning has been a prosperous research direction. This has spurred on research across multiple fronts, starting from gated recurrent models (Hochreiter & Schmidhuber, 1997; Cho et al., 2014), convolution (Bai et al., 2018a) to the recently popular self-attention based models (Vaswani et al., 2017).

The intrinsic hierarchical structure native to many forms of sequences have long fascinated and inspired many researchers (Socher et al., 2013; Bowman et al., 2014; 2016; Dyer et al., 2016). The study of recursive networks, popularized by (Socher et al., 2013) has provided a foundation for learning syntax-guided composition in language processing research. Along the same vein, (Tai et al., 2015) proposed Tree-LSTMs which guide LSTM composition with grammar. Recent attempts have been made to learn this process without guidance nor syntax-based supervision (Choi et al., 2018; Shen et al., 2017; Havrylov et al., 2019; Yogatama et al., 2016). Ordered Neuron LSTMs (Shen et al., 2018) proposed structured gating mechanisms, imbuing the recurrent unit with a tree-structured inductive bias. (Tran et al., 2018) shows that recurrence is important for modeling hierarchical structure. Notably, learning hierarchical representations across multiple time-scales (El Hihi & Bengio, 1996; Schmidhuber, 1992; Koutnik et al., 2014; Chung et al., 2016; Hafner et al., 2017) have also demonstrated reasonable success.

Learning an abstraction and controller over a base recurrent unit is also another compelling direction. First proposed by Fast Weights (Schmidhuber, 1992), several recent works explore this notion. HyperNetworks (Ha et al., 2016) learns to generate weights for another recurrent unit, i.e., a form of relaxed weight sharing. On the other hand, RCRN (Tay et al., 2018) explicitly parameterizes the gates of a RNN unit with other RNN units. Recent attempts to speed up the recurrent unit are also reminiscent of this particular notion (Bradbury et al., 2016; Lei et al., 2018).

The marriage of recursive and recurrent architectures is also notable. This direction is probably the closest relevance to our proposed method, although with vast differences. (Liu et al., 2014) proposed Recursive Recurrent Networks for machine translation which are concerned with the more traditional syntactic supervision concept of vanilla recursive nets. (Jacob et al., 2018) proposed RR-Net, which learns hierarchical structures on the fly. RR-Net proposes to learn to split or merge nodes at each time step, which makes it reminiscent of (Choi et al., 2018; Shen et al., 2018). (Alvarez-Melis & Jaakkola, 2016) proposed doubly recurrent decoders for tree-structured decoding. The core of their method is a depth and breath-wise recurrence which is similar to our model. However, METAGROSS is concerned with learning gating controllers which is different from the objective of decoding trees.

Our work combines the idea of external meta-controllers (Schmidhuber, 1992; Ha et al., 2016; Tay et al., 2018) with recursive architectures. In particular, our recursive parameterization is also a form of dynamic memory which gives our model improved expressiveness in similar spirit to memory-augmented recurrent models (Santoro et al., 2018; Graves et al., 2014; Tran et al., 2016).

## 5 CONCLUSION

We proposed Meta Gated Recursive Controller Units (METAGROSS) a sequence model characterized by recursive parameterization of gating functions. Our proposed method achieves very promising and competitive results on a spectrum of benchmarks across multiple modalities (e.g., language, logic, music). We propose a non-autoregressive variation of METAGROSS, which allows simple drop-in enhancement to state-of-the-art Transformers. We study and visualise our network as it learns a dynamic recursive parameterization, shedding light on the expressiveness and flexibility to learn dynamic parameter structures depending on the data.

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
