# OpenReview forum: "Metagross: Meta Gated Recursive Controller Units for Sequence Modeling"
_ICLR.cc/2020/Conference — Reject_

### Official Review · AnonReviewer3 · 2019-10-20
**Official Blind Review #3**

**Rating:** 3

**Review:**

The authors propose a variant of gating functions for recurrent neural networks and feed-forward layers of Transformer and apply it to variety of tasks including toy tasks such as sorting, tree traversal and more realistic tasks such as machine translation. The gating function is applied recursively for N number of steps and depth of recursion is learned softly in data-driven function. Authors show similar or slightly better performance of their approach when applied to LSTM and Transformer compared to vanilla LSTM and Transformer.

I have several comments regarding this work:

1) I believe there is a typo in equation in section 2 describing parametrization of o^{n}_{t} where h_{t-1} has un-necessary upper-script {n}

2) I would like to see the equations showing differences/similarities between MetaGross gating function and GRU/LSTM in Section 2 to better understand how MetaGross relates to previous work.

3) Calling parallel version of MetaGross that computes gating values given entire sequence *non-autoregressive* is really correct since decoding for machine translation is still done in autoregressive fashion one token at a time. I would suggest the authors to remove word non-autoregressive and just stick with word parallel.

4) The obvious connection with this work and work of Alex Graves on Adaptive Computation for Recurrent Neural Nets and many of its followups including Universal Transformers by Dehghani et al is missing from experimental comparison and is not even mentioned in related section. I believe that not mentioning these papers and not comparing to them empirically  this is a major drawback of this paper.

5) I don't think that Task 3 (Logical Inference) contains a language vocabulary of six words because it is a natural english language unless I am misunderstanding something.

6) What does EM stand for in Table 1. Would be great if you could include description of what EM, P(perplexity), n(depth of recursion) stands for in caption of Table 1

Overall it is a good and well written paper, although I believe that the variants of recursively gated functions have been proposed and applied before (see comment 4). Would be open to discussions and raising scores if authors convince me otherwise.

**Experience Assessment:**

I have published in this field for several years.

**Review Assessment: Checking Correctness Of Derivations And Theory:**

I assessed the sensibility of the derivations and theory.

**Review Assessment: Checking Correctness Of Experiments:**

I assessed the sensibility of the experiments.

**Review Assessment: Thoroughness In Paper Reading:**

I read the paper at least twice and used my best judgement in assessing the paper.

---

### Official Review · AnonReviewer1 · 2019-10-23
**Official Blind Review #1**

**Rating:** 3

**Review:**

This paper proposes a neural sequence modelling unit called METAGROSS.  In principle, the aim of this unit is to introduce recursive parametrization of  gating functions, building on the gated RNN paradigm.  The authors motivate this work by arguing that while gated-RNNs tackle vanishing gradient problems and facilitate learning long-range dependencies in sequences, improvements can be made with respect to learning on hierarchically-structured data.  The authors propose a method to do so by also learning the depth of the parametrization, and claim that the inductive bias that emerges from this configuration is beneficial to learning such tasks.

I think the idea behind this work is sensible: introducing a meta-controller with recursive parametrization of gating functions for hierarchical tasks is sensible.  Another also strong point of this paper is that several experiments under different settings are presented, along with ablation studies and some model exploration.  The authors also show that integrating a non-autoregressive variant of the proposed meta-controller into the Transformer architecture can also be beneficial.  Results in general do show improvement over compared architectures.

On the negative side and besides empirical/experimental evidence, the paper would be much more convinving with some more insight into the model itself and some more qualitative evidence.  Figure 1 shows the architecture of the proposed method with a max depth of 3 indicating soft recursion with grayscale levels.  However, this figure is not referenced in the text and not explained further.  The non-autoregressive version (sec 2.3) simply does away with the dependence on hidden states and applies the proposed architecture directly on the input.  I think it would be very beneficial to see a simple toy example where the benefits of utilizing this meta-architecture can be qualitatively explained.  Finally, one can argue that this work is incremental, in the sense that it is a (relatively straightfwd) combination of meta-controllers with recursive architectures.  Differences and variations with respect to other methods in literature should be more clearly explained.

some more questions
- although most experiments show some resilience with respect to the varying max depth parameter, have the authors noticed any problems and limitations arising from setting the max-depth to be very high?  for example, in Table 1 it seems that accuracy may drop when increasing max length.  I think that more discussions and ellaboration on this could be useful, as the authors also propose a way of learning the depth parameters and also state that this is task dependent.
- Figures 7-8-9 show variations of dynamic recursion on three databases.  Although these figures do show variation in learning (ranging from high activation fluctuations to static), it would be interesting to examine why this fluctuations occur in CIFAR
- which brings me to the second question:  although the results do not show this, could this task-dependent nature of the controller lead to more chances of overfitting on a given training set that may be noisy?
- Although the authors perform ablation tests with multiple units and evaluate for max depth, we don't see many experiments with depth of more than 2 or 3 (besides table 1 - in many experiments the max depth is not mentioned).  Are the conclusions the same with all experiments wrt depth?
- It would be useful to have a comment on model complexity

**Experience Assessment:**

I have read many papers in this area.

**Review Assessment: Checking Correctness Of Derivations And Theory:**

I assessed the sensibility of the derivations and theory.

**Review Assessment: Checking Correctness Of Experiments:**

I assessed the sensibility of the experiments.

**Review Assessment: Thoroughness In Paper Reading:**

I read the paper thoroughly.

---

### Official Review · AnonReviewer2 · 2019-10-24
**Official Blind Review #2**

**Rating:** 3

**Review:**

Update: As no rebuttal has been posted I stand by my assessment.

Summary
This papers proposes a recursive parameterization of gates in a recurrent model. Instead of directly conditioning gates on the input and previous hidden representation, the proposed model recursively calls itself to parameterize the gate. The recursion depth is dynamically determined (up to a predefined maximum recursion depth). The approach shows slight improvements over baselines on a range of tasks.

Strengths
Slight improvements on a wide range of downstream tasks
Qualitative analysis of dynamic recursion highlighting the adaptability of the method to different task properties

Weaknesses
From the equations in Section 2, I understand that a gate is conditioned not only on the input and previous hidden representation, but also the input and hidden representations from repeated application of the RNN cell for the given time step. As far as I understand, this is very closely related to ACT by Graves, Alex. "Adaptive computation time for recurrent neural networks." 2016. They also have a dynamic way of determining how many recursive RNN cell applications per step should be performed.
I am missing a clear description of differences of the proposed approach to the baselines tested in Section 3. At some point it is mentioned that a different optimizer was used (R-Adam). Are there any other confounding factors? I believe it is important to get clarity on these given that the difference between the model and baselines are very small.
Results in Table 1 are highlighted in a misleading way. For example, stacked BiLSTM do as well for tree traversal (EM, n=5 and EM, n=10). For logical inference, there is a more recent paper investigating the limits of RNNs and I believe comparisons on that dataset could strengthen the paper: Evans, Richard, et al. "Can Neural Networks Understand Logical Entailment?." ICLR 2018.

Minor Comments
p1: "the ability to reason deeply" – I understand you mean literally "deep", but also "reason" is a loaded term.
p1: "bears a totally different meaning" should be made concrete
p3: I am a bit confused by the fact that Metagross is used to extend Transformers here given that you called it a recurrent (and recursive) model on the previous slide.
p4: What is the number of layers used in he stacked BiLSTM and is this the same as the maximum depth used in Metagross? I believe for a fair comparison they should be the same. I am also missing stacked LSTM in experiments on logical inference (Table 2).
Questions to Authors

**Experience Assessment:**

I have read many papers in this area.

**Review Assessment: Checking Correctness Of Derivations And Theory:**

I assessed the sensibility of the derivations and theory.

**Review Assessment: Checking Correctness Of Experiments:**

I assessed the sensibility of the experiments.

**Review Assessment: Thoroughness In Paper Reading:**

I read the paper at least twice and used my best judgement in assessing the paper.

---

### Decision · Program_Chairs · 2019-12-19

**Decision:**

Reject

**Comment:**

This paper proposes a recurrent architecture based on a recursive gating mechanism. The reviewers leaned towards rejection on the basis of questions regarding novelty, analysis, and the experimental setting. Surprisingly, the authors chose not to engage in discussion, as all reviewers seems pretty open to having their minds changed. If none of the reviewers will champion the paper, and the authors cannot be bothered to champion their own work, I see no reason to recommend acceptance.